# Molecular Insight into the Therapeutic Effects of Stem Cell-Derived Exosomes in Respiratory Diseases and the Potential for Pulmonary Delivery

**DOI:** 10.3390/ijms23116273

**Published:** 2022-06-03

**Authors:** Mohammad H. Azhdari, Nima Goodarzi, Mohammad Doroudian, Ronan MacLoughlin

**Affiliations:** 1Department of Cell and Molecular Sciences, Faculty of Biological Sciences, Kharazmi University, Tehran 15719-14911, Iran; azhdari.mh99@gmail.com (M.H.A.); nima.goodarzi99@gmail.com (N.G.); 2Research and Development, Science and Emerging Technologies, Aerogen Limited, IDA Business Park, H91 HE94 Galway, Ireland; 3School of Pharmacy, Royal College of Surgeons, D02 YN77 Dublin, Ireland; 4School of Pharmacy and Pharmaceutical Sciences, Trinity College Dublin, D02 PN40 Dublin, Ireland

**Keywords:** extracellular vesicles, nanosized vesicles, lung diseases, stem cell-derived exosomes

## Abstract

Respiratory diseases are the cause of millions of deaths annually around the world. Despite the recent growth of our understanding of underlying mechanisms contributing to the pathogenesis of lung diseases, most therapeutic approaches are still limited to symptomatic treatments and therapies that only delay disease progression. Several clinical and preclinical studies have suggested stem cell (SC) therapy as a promising approach for treating various lung diseases. However, challenges such as the potential tumorigenicity, the low survival rate of the SCs in the recipient body, and difficulties in cell culturing and storage have limited the applicability of SC therapy. SC-derived extracellular vesicles (SC-EVs), particularly SC-derived exosomes (SC-Exos), exhibit most therapeutic properties of stem cells without their potential drawbacks. Similar to SCs, SC-Exos exhibit immunomodulatory, anti-inflammatory, and antifibrotic properties with the potential to be employed in the treatment of various inflammatory and chronic respiratory diseases. Furthermore, recent studies have demonstrated that the microRNA (miRNA) content of SC-Exos may play a crucial role in the therapeutic potential of these exosomes. Several studies have investigated the administration of SC-Exos via the pulmonary route, and techniques for SCs and SC-Exos delivery to the lungs by intratracheal instillation or inhalation have been developed. Here, we review the literature discussing the therapeutic effects of SC-Exos against respiratory diseases and advances in the pulmonary route of delivery of these exosomes to the damaged tissues.

## 1. Introduction

Respiratory diseases are among the top causes of death globally. It was estimated that cumulatively, respiratory diseases were the third largest cause of death and were responsible for approximately 4 million deaths worldwide in 2019 [1]. Novel approaches, such as nanomedicine and regenerative medicine, could be beneficial in the diagnosis or treatment of various lung disorders [2,3,4,5]. Despite advances in modern medicine and drug design techniques, no drug has been developed to cure major respiratory diseases. Most available drugs mainly help control and restrict the progress of the disease or prevent complications. Cell-based therapies hold great promise for treating various lung diseases. In cell therapy, viable cells are introduced into the body via different routes to exert their particular therapeutic effects. Stem cells (SCs) have been found to have protective and regenerative properties on different tissues, and numerous studies have investigated the advantages of SCs in the treatment of disorders. SCs can be obtained from four different sources: embryonic tissues, fetal tissues, adult tissues (e.g., mesenchymal stem cells (MSCs) derived from different tissues), and differentiated somatic cells resulting from genetic manipulation, known as induced pluripotent SCs (iPSCs) [6]. The administration of SCs has been shown to be safe and effective in various respiratory diseases, such as chronic obstructive pulmonary disease (COPD), acute respiratory distress syndrome (ARDS), and pulmonary fibrosis (PF) [7]. However, the therapeutic use of SCs has its downsides: (I) SCs secrete varied growth factors that may have the potential to induce tumors; (II) the survival rate of SCs after administration is low; (III) SCs may block some small-diameter pulmonary blood vessels due to their size, and (IV) specific requirements make it challenging to culture and store SCs [8].

SC-derived exosomes (SC-Exos) as a cell-free therapy have been shown to be a promising strategy in addressing challenges associated with the administration of SCs, according to their negligible tumorigenicity, easier manipulation, standard storage requirements, and availability [9]. Exosomes are the smallest type of lipid-bilayer EVs with a 30–120 nm diameter size, similar to other EVs (microvesicles and apoptotic bodies), take part in intracellular communications as a paracrine factor, and can be secreted from many cell types including SCs [10,11]. Although exosomes secreted from different sources share a conserved general composition (Figure 1), they can be wildly diverse depending on the origin of the cell type and its microenvironmental stresses and conditions [12]. Accordingly, by inheriting their origin cell molecular pattern in their cargo composition, SC-Exos can affect their recipient cells similarly to their parental cells [13].

The administration route is one of the determining factors in the success and effectiveness of therapeutic agents. Introducing drugs directly to the lungs is suitable for treating respiratory diseases, mainly because the therapeutic agent could reach the region affected by the disease without entering circulation [14]. Therapeutic ingredients can be injected directly into the trachea (i.e., intratracheal instillation) or introduced to the lungs in aerosolized form using devices such as nebulizers and inhalers. Instillation via an endotracheal tube temporarily placed in the trachea is more invasive than inhalation; however, both approaches are considered noninvasive compared to invasive methods such as parenteral administration [15,16,17]. Several clinical and preclinical studies have investigated the potential of pulmonary delivery of SC-EVs and SC-Exos via intratracheal instillation and inhalation. Further in this review, we discuss the biogenesis of SC-Exos, their role and mechanism of action in treating lung diseases, and their potential to be administered via the direct pulmonary route.

## 2. Exosome Biogenesis

Exosome biogenesis starts with the formation of an early endosome that matures into a late endosome. Exosomal cargos are then segregated into microdomains on the late endosome membrane. Exosome biogenesis starts with the formation of an early endosome that forms by several endocytic vesicle fusion. Exosomal cargos are then segregated into microdomains on the endosome membrane. Further, these microdomains bud into the lumen of the endosome and form a multivesicular body (MVB), also referred to as a late endosome, containing numerous intraluminal vesicles (ILVs) that can be secreted upon fusion of this MVB with the plasma membrane or become degraded by MVB fusion with the lysosome [18]. Although this pathway is generally accepted, it does not seem to be the only mechanism of exosome biogenesis. Multiple studies suggested that exosomes can be formed and secreted by direct budding from the plasma membrane in some cell types and circumstances [19,20,21].

The protein sorting process into ILVs can be regulated by both endosomal sorting complexes required for transport (ESCRT)-dependent and independent pathways. ESCRT machinery consists of four main ESCRT complexes, including ESCRT-0, ESCRT-I, ESCRT-II, and ESCRT-III, and sort proteins into ILVs in ubiquitin-dependent and independent pathways. In the ubiquitin-dependent pathway, ubiquitylated proteins are recognized and encountered by ESCRT-0. By joining ESCRT-I and ESCRT-II to the complex, a strong recognition domain with a high affinity for ubiquitinated substrates forms on the endosomal membrane and buds into the inside. With ESCRT-III convergence with the complex, these buds can further be released into the intraluminal space, and the cargo proteins can be deubiquitinated by the deubiquitylating enzyme to rescue the ILVs from lysosomal degradation. In addition, nonubiquitinated proteins can be sorted into ILVs via the ESCRT pathway intersection by syntenin and ALIX (ALG-2-interacting protein X; an ESCRT accessory protein) proteins, which clusters nonubiquitinated proteins into microdomains and bridges them and the Vps32 (Vacuolar sorting-associated protein 32) ESCRT-III subunit, respectively [22]. Additionally, ESCRT-independent pathways mediated by ceramide (ESCRT-independent ILV budding), tetraspanins such as CD63, CD81, CD82, and CD9 (ESCRT-independent endosomal sorting regulation), and HSP70 and HSC70 (selective cytosolic protein cosorting into ILVs) can result in ILV formation and regulation separately from the ESCRT machinery. However, the role of both the ESCRT-dependent and independent pathway in determining ILV composition is necessary and routinely regulates ILV formation simultaneously [23].

## 3. Stem Cell-Derived Exosome Therapeutic Effect on Lung Diseases

### 3.1. Acute Respiratory Distress Syndrome

Acute respiratory distress syndrome (ARDS) is a clinical condition resulting from various risk factors, including pneumonia, sepsis, aspiration of gastric contents, trauma, and other less common precipitants [24,25]. ARDS is considered a more severe form of acute lung injury as it shares similar pathophysiological conditions. ARDS is associated with increased alveolar–capillary permeability to fluid, proteins, neutrophils, and inflammatory cells resulting in enhanced inflammation, accelerated autophagy, epithelial-mesenchymal transition (EMT), and decreased alveolar fluid clearance, leading to pulmonary edema, impaired gas exchange, hypoxemia, and PF [26,27,28]. Various stimuli such as hypoxia, cytokines, chemokines, thrombins, primed leukocytes, lipopolysaccharides (LPS), and damage-associated molecular patterns can shift the alveolar-capillary membrane toward dysfunction [24]. With the relatively high rate of ARDS-associated mortality and morbidity [28], and, on the other hand, the promising SC and SC-Exo regenerative and immunomodulatory properties, several studies evaluated SC-Exo effects on ARDS models and investigated the underlying mechanisms (Table 1) [29,30,31,32,33,34].

Various in vitro and in vivo studies on LPS-induced ARDS models reported a significant increase in inflammatory factors, leading to enhanced autophagy and EMT progress. Investigations on an LPS-induced murine lung epithelial-12 (MLE-12) model revealed that upregulation in the nuclear factor kappa-light-chain-enhancer of activated B cells (NF-κB) signaling pathway is the main factor in LPS-induced inflammation, which further initiates EMT by the hedgehog signaling pathway activation via sonic hedgehog protein transcription. Additional treatment with MSC-Exos suggests NF-κB regulation, leading to inflammation suppression and EMT reverse. Further investigations into the underlying MSC-Exo mechanism of action demonstrate that MSC-Exo contains miR-182-5p and miR-23a-3p, targeting *Ikbkb* and *Usp5* genes, respectively. *Ikbkb* and *Usp5* encode IKKβ (inhibitor of nuclear factor kappa-B kinase subunit beta) and Usp5 (ubiquitin-specific peptidase 5) proteins, and their silencing causes a decrease in IKKβ expression and an increase in its ubiquitination, leading to a coinhibition of the NF-κB pathway [35]. A study on LPS-induced NR8383 cell treatment with UCMSC-Exos demonstrated that UCMSC-Exos can reduce the inflammatory reaction, suppress oxidative stress response, and decrease NF-κB expression via frizzled class receptor 6 (FZD6) downregulation by transferring miR-22-3p into LPS-induced cells and targeting FZD6 directly [37]. Although FZD6 mainly serves as an initiator in the Wnt signaling pathway, it can lead to NF-κB activation via Wnt/β-Catenin signaling pathway mediation [44]. Serum amyloid A-3 (SAA3) is a downstream gene in the NF-κB signaling pathway whose activation enhances proinflammatory cytokine activation [45], and its upregulation has been shown to play a role in inflammation occurrence in MLE-12 cells after LPS treatment. It is shown that MSC-derived exosomal miR-30b-3p is another anti-inflammatory factor in MSC-Exos that can suppress inflammation and enhance cell proliferation by decreasing SAA3 expression [39]. In addition, it is reported that MSC-Exos inhibit inflammation in an intestinal ischemia reperfusion-induced (IIR-induced) ARDS model by NF-κB suppression via downregulating toll-like receptor-4 (TLR4); however, the underlying mechanism has not yet been identified [42].

In addition to NF-κB, MSC-Exos are reported to ameliorate ARDS by manipulating other signaling pathways in different ARDS models. A recent study on an LPS-induced mice ARDS model suggested that hUCMSC-Exos treatment can effectively reverse LPS induction complications, including inflammatory infiltration, increased bronchoalveolar lavage fluid protein concentration, and increased interleukin (IL)-6 and IL-1β concentrations. Moreover, it was revealed that the administration of hUCMSC-Exos significantly enhances autophagy. Further investigations on LPS-treated human pulmonary alveolar epithelial cells and the underlying mechanism of hUCMSC-Exos in autophagy induction stand for the exosomal miR-377-3p pivotal role, which targets the regulatory associated protein of mTOR (RPTOR). Although the exact mechanism by which RPTOR silencing induces autophagy is still unclear, it is shown that exosomal miR-377-3p-mediated RPTOR downregulation can effectively improve LPS-induced ARDS in vitro and in vivo [36]. Figure 2 demonstrates the underlying mechanism of actions revealed by the mentioned studies in which SC-Exos improve alveolar cells in ARDS.

As one of the pivotal players in ARDS occurrence and progression, alveolar macrophages can also be affected by MSC-Exos, resulting in ARDS improvement. It is demonstrated that bone marrow-derived MSC (BMSC)-derived exosomes (BMSC-Exos) uptake by alveolar macrophages in LPS-induced ARDS mice can induce their differentiation from a proinflammatory type into an anti-inflammatory type via inhibiting M1 polarization and promoting M2 polarization [46]. Further, it is shown that by targeting and suppressing Beclin-1 mediated by exosomal miR-384-5p, BMSC-Exo treatment can inhibit the viability loss, prevent apoptosis, and attenuate autolysosome and autophagosome punctum formation in LPS-induced alveolar macrophages, which leads to controlling the alveolar macrophage autophagy induced by LPS [43].

### 3.2. Bronchopulmonary Dysplasia

Bronchopulmonary dysplasia (BPD) is a multifunctional chronic lung disease associated with prematurity and is one of the most common complications in preterm infants with low birth weight [47]. BPD can be associated with airway distortion, bronchial muscle hypertrophy, pulmonary hypertension leading to capillary remodeling, and severe inflammation with extensive fibrotic changes, which altogether can cause impaired pulmonary function and reduced exercise tolerance that continues into adulthood [48,49]. The improvement in infant viability in past decades has resulted in more premature infants with diagnosed BPD surviving into adulthood, and, as a consequence, more adult cases with BPD are diagnosed [50]. Therefore, new and effective treatments are required for treating and reversing BPD complications, and MSC-Exos are shown to be promising to address this issue. It is shown that exposing newborn mice to hyperoxia (HYRX; 75% O_2_) from postnatal day (PN) 1 to PN7 and returning them to room air (NRMX) for the next seven days demonstrates similar histological patterns to the human BPD, characterized by alveolar growth impairment, large airspaces, and incomplete alveolar septation. A more severe model can be achieved by exposing newborn mice for an extra seven days (from PN1 to PN14), resulting in additional collagen deposition. These animal models and MSC-Exos from human bone marrow and the umbilical cord Wharton’s Jelly effects on them were investigated in a study in the short- and long-term. It has been shown that treatment of the HYRX-exposed group with a single dose of intravenous injection of BMSC-Exos or Wharton’s Jelly—a mucoid connective tissue—MSC (WJMSC)-derived exosomes (WJMSC-Exos) at PN4 can dramatically improve the alveolarization and restore lung architecture almost completely in the short-term (PN14) compared with the untreated HYRX-exposed group. This effect was compared with human dermal fibroblast-derived exosomes as a control, which demonstrated no significant protective effect as assessed at PN14. Over a long-term assessment, these effects were studied at PN42, suggesting robust improvement in pulmonary development, significantly decreased collagen deposition, peripheral pulmonary arterial remodeling, HYRX-induced PH improvement, and overall lung function improvement after HYRX-induced lung injury. Additional whole-organ RNA sequencing and a gene ontology analysis on the NRMX- and HYRX-exposed groups at PN7 indicate an upregulation in genes associated with adaptive immune response, inflammatory response, and leukocyte-mediated immunity in the HYRX-exposed group. This HYRX-mediated immune response induction is demonstrated to be modulated in the WJMSC-Exo-treated group by blunting genes involved in inflammation, adaptive immune response, IFN-γ-mediated signaling, and production of cytokines and granulocytes leading to suppressing the M1 (proinflammatory phenotype) macrophage and augmenting the M2-like (anti-inflammatory phenotype) macrophage and shifting the overall transcriptome profile of the HYRX group toward the NRMX group [51]. Another study compared the effect of “early” and “late” intervention of WJMSC-Exos on prolonged neonatal HYRX-induced lung injury. In this study, newborn mice were exposed to HYRX (75% O_2_) for 14 days and divided the animals into two groups, treated with a single dose of MSC-Exos at PN4 (early intervention) or a serial dose of MSC-Exos at PN18, 25, 32, and 35 (late intervention). It is demonstrated that the late intervention of MSC-Exos can dramatically improve alveolarization, reverse elevations in pulmonary vascular muscularization, ameliorate alterations in the right ventricular hypertrophy, and restore functional exercise capacity in HYRX-exposed mice at PN60, with no substantial difference with the early intervention [52]. Studies on the MSC-Exos mechanism of action suggest tumor necrosis factor-inducible gene 6 protein (TSG-6), an immunomodulatory protein, as a crucial factor in MSC-Exo effects on BPD models. After validating the human WJMSC-Exo immunomodulatory and regenerative effect on BPD models, BPD models and human tracheal aspirates in patients with BPD development were checked for TSG-6 RNA level change, demonstrating a significant elevation in the expression of TSG-6 in comparison with the non-BPD group. However, in the WJMSC-Exo-treated group, the TSG-6 level decreased to that of the NRMX group. By Western blotting, TSG-6 was detected in WJMSC-Exos. Human recombinant TSG-6 was injected intraperitoneally into PBD mouse models at PN2 and 4 to investigate whether TSG-6 in WJMSC-Exos acts as one of the mediators in improving the BPD models, which results in significant immunomodulation and an overall lung architecture improvement. Additionally, loss of function of TSG-6 was performed by the administration of the TSG-6 neutralizing antibody or the introduction of a TSG-6 short interfering RNA (siRNA) into exosome donor cells (25 weeks gestational age human WJMSC). A further analysis suggests no difference between BPD and the exosome-treated group, demonstrating TSG-6 plays a crucial role in the WJMSC-Exo effect on attenuating BPD in mouse models [53].

### 3.3. Pulmonary Hypertension

Pulmonary hypertension (PH) is a chronic disorder defined as an increase in mean pulmonary arterial pressure (greater than 25 mm Hg at rest). PH has been divided into five subtypes: (I) pulmonary arterial hypertension (PAH); (II) PH caused by left-sided heart disease; (III) PH due to lung disease and/or hypoxia; (IV) chronic thromboembolic PH; and (V) PH with unclear or multifactorial etiologies [54]. Several factors can induce PH, and currently, most therapies focus on improving the symptoms of PH. In recent years, advances in the understanding of underlying molecular mechanisms of PH, such as vascular remodeling and the role of leukocytes, led to the development of new therapies that potentially can reverse the underlying mechanisms of the disease. SCs and their secretome have been proven to reverse the PH process by simultaneously affecting several pathways [55].

Recent studies indicated that SC-Exos are effective in PH treatment. A 2012 study demonstrated that MSC-Exos and human umbilical cord-derived MSC (UCMSC)-derived exosomes (hUCMSC-Exos) inhibit PH through several mechanisms. The injection of MSC-conditioned media into the hypoxia-induced PH mouse model suppressed the increase in the hypoxia-induced mitogenic factor and monocyte chemoattractant protein-1. Further experiments showed that MSC-conditioned, which is depleted from exosomes, had no considerable inhibitory effect, but isolated MSE-derived exosomes suppressed hypoxic inflammation in the animal model. MSC-Exos exert their function by inhibiting signal transducer and the activator of transcription 3 (STAT3); its activation has a crucial role in respiratory epithelial inflammatory responses. In addition, MSC-Exos affect levels of several types of miRNA by suppressing STAT3; the transcription of the miR-17 superfamily is upregulated in pulmonary artery endothelial cells in response to hypoxic conditions. Additionally, the miR-204 level in pulmonary artery smooth muscle cells (PASMCs) is inversely related to the severity of PH. The study suggests that MSC-Exos might attenuate PH by preventing the hypoxic induction of the miR17 superfamily and enhancing the levels of miR-204 [55,56,57].

PH, particularly PAH, is mainly associated with a progressive remodeling of small pulmonary arteries, causing right heart failure and death [58]. Studies indicated that dysregulation of some signaling pathways, such as the Wnt pathway, could involve PAH pathogenesis [59,60]. Wnt5a is a signaling protein that activates the Wnt pathway and regulates cell fate and embryo development. The reduction of Wnt5a has been found as a role player in regulating PAH through vascular remodeling [61]. Zhang et al., demonstrated that MSC-Exos are effective against vascular remodeling and right ventricular hypertrophy caused by monocrotaline-induced PH by regulating Wnt5a. The study also showed that the mechanism of action of MSC-Exos might be related to controlling the bone morphogenetic protein type II receptor (BMPRII). Mutations in the BMPRII gene, as a transmembrane receptor associated with the bone morphogenetic protein pathway, are related to PAH’s pathogenesis. Furthermore, MSC-Exos prevented the endothelial to mesenchymal transition process by regulating the Wnt5a/BMPRII signaling pathway [62,63,64].

Mitochondrial dysfunction is another factor involved in the pathogenesis of PH [65,66]. Sirtuin 4 (SIRT4), a mitochondrial protein, is a member of the mammalian sirtuin family with ADP-ribosyltransferase activity. In response to hypoxic conditions, the expression of the SIRT4 gene increases to block the tricarboxylic acid cycle and protect the cell against oxygen deficiency. However, chronic activation of SIRT4 might lead to mitochondrial dysfunction [67,68]. Hogan et al., indicated that MSC-Exos could decrease SIRT4 gene expression in hypoxia-exposed PASMCs and the semaxinib/hypoxia rat model of PAH. It is proposed that reducing SIRT4 increases tricarboxylic acid cycle metabolites and, consequently, mitochondrial metabolism [69].

### 3.4. Pulmonary Fibrosis

Pulmonary fibrosis (PF) is a progressive and destructive lung condition driven by the fibrotic response due to various factors. Idiopathic PF (IPF), the most common form of PF, is identified by replacing healthy lung tissue with extracellular matrix and transforming alveolar structure [70,71]. Currently, there are two to drugs approved by the Food and Drug Administration (FDA) for PF, but neither of them can reverse the disease process, and the only available cure is lung transplantation, which has its own risks. According to studies, dysregulation of hedgehog, transforming growth factor beta (TGF-β), notch and fibroblast growth factor signaling pathways associate lung development and repair. However, dysregulation of these pathways can lead to different diseases such as PF [72]. BMSC-Exos have been shown to attenuate PF in the rat model of silica-induced PF. BMSC-Exos can reduce levels of the TGF-β1 and Wnt/β-catenin signaling pathways and inhibit the progress of the EMT process [73].

In recent years, the understanding of the function of miRNA in IPF and related pathways has increased. An alteration of about 10 percent of miRNA in an IPF lung indicates that miRNAs might play essential roles in treating PF. Previous studies demonstrated the role of some types of miRNA, such as let-7, miR-155, miR-21, and miR-29, in the pathogenesis of IPF [74]. Xu et al., showed that human UCMSC-Exos could transfer let-7i-5p into fibroblast cells and interfere with TGFBR1/Smad3 signaling pathways. The effect of UCMSC-Exos against the silica-induced mice model was more beneficial than MRC-5 (human embryonic lung fibroblasts)-derived exosomes (Figure 3a) [75]. Sun et al., demonstrated that menstrual blood-derived SC-derived exosomes contain Let-7 miRNA and relieved bleomycin-induced PF and alveolar epithelial cell damage (Figure 3b). Exosomal Let-7 suppresses the expression of lectin-like oxidized low-density lipoprotein scavenger receptor-1 (LOX1) and has a protective effect on alveolar epithelial cells [76]. Moreover, the Let-7 family and two other miRNAs, miR-30a and miR-99, have been identified in lung spheroid cell-derived exosomes (LSC-Exos). LSCs are derived from adult lungs and contain lung progenitor cells and supporting stromal cells and can be used for therapeutic applications. It was indicated that LSC-Exos, compared with MSC-Exos, have a better performance against the rat model of bleomycin- and silica-induced PF [77,78].

The immune system is another factor associated with the pathogenesis of IPF [79]. It is well known that SC-Exos have immunomodulatory properties, and recent studies showed that at least part of the protective effects of SC-Exos against PF is due to the immune system’s regulation [8]. SC-Exos are reported to regulate the function and behavior of different immune system cell types, such as neutrophil, macrophage, and T cells, and their immunomodulatory properties are effective in IPF treatment [80]. Mansouri et al., showed that MSC-Exos could inhibit and revert the process of bleomycin-induced PF and suggested that this effect may be due to the systemic modulation of monocyte phenotypes [81]. Furthermore, iPSC-derived exosomes (iPSC-Exos) can alleviate PF by suppressing M2-type macrophages. A recent study showed that iPSC-Exos contain high levels of MiR-302a-3p miRNA and detected the binding between MiR-302a-3p and ten-eleven translocation 1 (TET1). TET1 functions as a DNA methylation and gene transcription modulator and may have roles in fibrosis by regulating M2-type macrophages. The study suggests that MiR-302a-3p is a critical mediator in iPSC-Exos that ameliorates PF by TET1 and, consequently, M2-type macrophages [82].

### 3.5. Asthma

Asthma is one of the most common chronic and nontransmissible lung diseases affecting children and adults. Asthma has been regarded as a heterogeneous condition caused by a complex interplay of genetic and environmental factors. Inhaled corticosteroids remain the mainstay intervention to control asthma, but their effects vary based on the subtype of disease [83,84]. SCs and SC-Exos may be a potential therapy that may use as a complement or alternative medication [85]. Immune system malfunction and disruption in signaling pathways have been associated with the pathophysiology of asthma [86,87], and SC-Exos may be beneficial by regulating the involved pathways [88]. Dong et al., demonstrated that MSC-Exos could reduce lung inflammation and pulmonary hyper-responsiveness in the mice model of severe steroid-resistant asthma. Experiments on LPS-stimulated RAW 264.7 cells showed that MSC-Exos regulate macrophage polarization activation of the NF-κB and PI3K/AKT signaling pathways through tumor necrosis factor receptor-associated factor 1 protein [89]. Moreover, another study found that MSC-Exos could moderately increase lung interstitial macrophages and promote protective effects in ovalbumin-sensitized mice. Interstitial macrophages are shown to produce IL-10 and improve allergic asthma in mice [90]. The miRNA content also plays some roles in the function of exosomes against asthma. A study identified miR-301a-3p in adipose-derived MSC (ASC)-derived exosomes (ASC-Exos) and tested its function against platelet-derived growth factor-treated airway smooth muscle cells (ASMCs). The results imply that the exosomal miR-301a-3p reduces remodeling and inflammation of PDGF-BB-treated ASMCs by targeting the 3ʹUTR region of STAT3 [90].

### 3.6. Other Pulmonary Disorders

Sepsis is a life-threatening condition caused by an improper host response to infection. The lung is the most common site of the infection and may fail during sepsis [91]. Recent studies proposed some roles for SC-Exos in improving sepsis-induced lung failure. SC-Exos could modulate various pathways involved in sepsis-induced lung injury pathogenesis and regulate the immune system. IL-27 is a cytokine that increases during sepsis, and its suppression could be a potential target in treating sepsis [92]. ASC-Exos have been shown to attenuate lung injury in septic mice by decreasing pulmonary macrophages and reducing their IL-27 expression [93]. The molecular mechanism of action of SC-Exos could be related to their miRNA content. A study identified exosomal miR-16-5p derived from ASC as a regulator of TLR4 in septic mice. By this mechanism, ASC-Exos promoted macrophage polarization and alleviated sepsis-induced lung injury [93].

SC-Exos have been shown to be effective against cigarette smoke (CS)-induced lung inflammation. Xu et al., demonstrated that BMSC-Exos could regulate inflammatory and apoptosis-related factors by repressing the HMGB1/NF-κB pathways. Regulatory effects of exosomes reduced CS-induced lung injury in vivo and in vitro [94]. In another study, immunomodulatory effects of ASC-Exos against CS-induced lung injury were determined. The results suggest that alveolar macrophage pyroptosis is associated with CS-induced lung inflammation, and ASC-Exos ameliorate the inflammation and mucus hypersecretion in CS-exposed mice by preventing the pyroptosis process in alveolar macrophages [95]. Some recent studies reported the benefits of SC-Exos to treat other pulmonary conditions, such as ventilator-induced lung injury [95], diffuse alveolar hemorrhage [96], ischemia/reperfusion injury (IRI) [97], pulmonary embolism [98], and basement membrane-induced fibrosis [99].

## 4. Pulmonary Delivery of Stem Cell-Derived Exosomes

Several routes may be used to deliver drugs into the body, and preferring one depends on multiple factors such as the disease, the affected organ, and the drug’s properties [100]. The pulmonary route has been used for thousands of years and has several advantages over other delivery routes: The lungs have a large surface area that is suitable for the rapid absorption of drugs; there are no extreme PH and distractive enzymes that exist in other routes; drugs do not need to pass first-pass liver; drugs rapidly transfer from the alveolar area to the lungs, and, finally, delivering drugs directly to the lungs when the lungs are the disease site minimizes the side effects on other organs [101,102,103]. However, the lung has its defense mechanism; for instance, macrophages and mucus capture and remove foreign particles in the airway, and to deliver therapeutic agents, this removal by macrophages should be taken into account [104]. Nevertheless, the pulmonary route is appropriate for drug delivery and is used for various respiratory and lung diseases such as asthma [105]. Intratracheal instillation and inhalation are two noninvasive pulmonary drug delivery approaches recently used to introduce SC-Exos to the lungs.

### 4.1. Intratracheal Instillation

The intratracheal instillation process can expose chemical substances directly to the trachea [15]. Instillation has some advantages over inhalation; compared to inhalation, installation is a simple technique, allows precise determination of delivered dose to the lungs, prevents the exposure of administered substances to the skin, and permits introducing substances to a specific region in the lungs. However, the use of this method is limited mainly because of its invasive nature, uneven distribution, and prolonged clearance of substances in the lungs (Figure 4) [106]. Instillation conventionally has been used to evaluate the respiratory toxicity of substances and the induction of pulmonary inflammation in animal models, but many studies employed this method to administer therapeutic agents [107]. A few studies investigated the effects of pulmonary instillation of SC-Exos on animal models of some lung disorders. Table 2. summarizes some of these studies.

### 4.2. Inhalation

Inhalation therapy has become more prevalent in recent years as it offers a noninvasive and efficient drug delivery route to the lungs. For a therapeutic agent to be inhaled, in combination with the aerosol generator of choice, it should form an aerosol droplet within a highly respirable distribution, usually considered to be between 1 and 5 μm (Andrersen Cascade Impactor, operated at 28.3 liters per minute). Additionally, aerosol generator devices are medical devices that aerosolize drugs to deliver them via inhalation [109]. Currently, there are three prevalent aerosol generator technologies in commercial use, with many new technologies under development or as of yet not commercially available: (I) pressurized metered-dose inhalers (pMDIs); (II) dry powder inhalers; and (III) medical nebulizers [110]. The choice of device should consider the required target lung dose and the likely patient intervention at the time of administration, as both have been shown to have significant influence [111,112]. The pMDIs are the most prescribed device for lung diseases such as COPD and asthma, but regarding clinical research, nebulizers are the most used instrument [113]. Nebulizers are preferred for early experiments because formulating a liquid is more accessible than making dry powder, and also nebulizers are appropriate for use on animal models [114]. Among the three common types of nebulizers—jet, ultrasonic, and vibrating mesh nebulizers (VMNs)—the jet nebulizers have been the most commonly used [115]. Some studies investigated jet and VMNs to deliver SCs into the lungs. Ultrasonic nebulizers are not considered appropriate for SC delivery, primarily because of the excessive heat produced and its effect on SC viability [116]. A 2013 study showed that, among three nebulizers, MSCs had the highest survival rate when jet nebulizers were used as the delivery device; however, aerosol generator technologies continue to advance, and so they warrant further investigations [117,118,119].

A few studies have investigated the effects of nebulization on SC-secretome or exosomes and its application in pulmonary diseases. The VMN was shown to be a suitable device to deliver BMSC and UCMSC secretome to the lungs [116]. Moreover, Dinh et al., used a jet nebulizer to introduce MSC- and LSC-Exos to the silica model of induced PF. Approximately 10 × 10^9^ exosome particles per kg of body weight were given through the nebulizer, and any adverse effects were observed on the heart, kidneys, liver, and spleen after treatment. Interestingly, nebulized secretomes and exosomes alleviated PF in silica-induced models and improved lung health (Figure 5) [77]. Inhalation delivery of exosomes is being investigated in several recent clinical trials. The following section will mainly focus on the role of exosomes and their delivery route in these trials.

## 5. Clinical Trials

Some clinical trials have been conducted to evaluate the benefits of SC-Exos against lung disorders. Coronavirus disease 2019 (COVID-19) is an infectious disease caused by severe acute respiratory syndrome coronavirus 2 (SARS-CoV-2), a novel coronavirus that was first isolated from the respiratory epithelium of patients suffering from unexplained pneumonia in 2019 [120,121]. COVID-19 affects different organs and causes serious complications, such as pneumonia, ARDS, acute liver injury, and multiorgan failure [122,123]. SCs and SC-Exos, with their immunomodulatory and protective properties, can help control and prevent destructive COVID-19 complications [8]. Currently, a number of clinical and preclinical trials are underway in order to evaluate the safety and efficacy of SCs and SC-Exos in severe COVID-19 patients. Table 3 indicates some of the clinical trials that assessed the safety and effectiveness of SC-Exos against respiratory diseases.

According to the ongoing and completed trials, inhalation is the preferred route in clinical studies (rows 1–6 in Table 3). In one study, preclinical and clinical efficacy and safety of MSC-derived EVs were determined. Nebulizing human ASC-derived EVs (haMSC-EVs) using VMNs showed an 80 percent increase in survival rate in *P. aeruginosa*-induced murine lung injury model. Considerable side effects were not observed after administering haMSC-EVs nebulization to twenty-four volunteers. Based on these results, a randomized, double-blind, placebo-controlled clinical study is in progress to determine the efficacy of haMSC-EV nebulizers for treating ARDS (NCT04313647) [124]. In another nonrandomized open-label clinical trial, 24 COVID-19 patients were treated with exosomes (ExoFlo^TM^) derived from allogeneic BMSC. At the end of the study, the intravenous administration of exosomes led to a survival rate of 83 percent. In addition, ExoFlo^TM^ improved oxygenation, reduced cytokine storm, and regulated the immune system [125]. Future clinical trials are needed to determine all aspects of SC-Exos in the treatment of respiratory diseases.

## 6. Conclusions and Future Perspective

Cell-free therapy has gained much momentum and focus in recent years, and numerous studies have investigated the therapeutic potential of exosomes. In this review, we discussed the clinical and preclinical studies around SC-Exos for improving the condition of patients suffering from lung diseases. Further, we talked about the potential of pulmonary drug delivery as a promising route for administering SC-Exos. Although our knowledge of exosomes has increased steadily over recent years, the exact mechanism of action of exosomes remains to be elucidated. Recent studies attributed the function of exosomes to their RNA cargos, but the precise role of other components is not clearly understood. In addition, limitations such as high costs and technical challenges make it difficult to isolate and purify considerable amounts of particular exosomes; however, recent progress in detecting exosomal bioactive molecules responsible for their therapeutic effects can lead to designing exosome-mimic particles with similar therapeutic effects. Another challenge is to immortalize SCs in order to obtain excess quantities of exosomes. However, the cell should be manipulated with its risk and limitations to achieve this goal. Further studies should be carried out to determine the potential of aerosolized SC-Exos and identify the proper device to administer them.

## Figures and Tables

**Figure 1 ijms-23-06273-f001:**
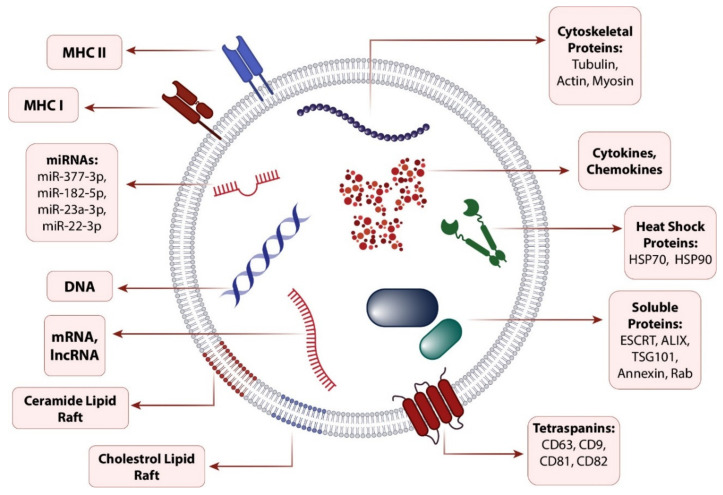
Exosome composition. Exosomes share a conserved cargo composition consisting of proteins, DNA, mRNAs, and miRNAs, which can be regulated according to the secreting cell type and its microenvironment. Abbreviations: MHC, Major histocompatibility complex; miRNA, microRNAs; lncRNA, long noncoding RNA.

**Figure 2 ijms-23-06273-f002:**
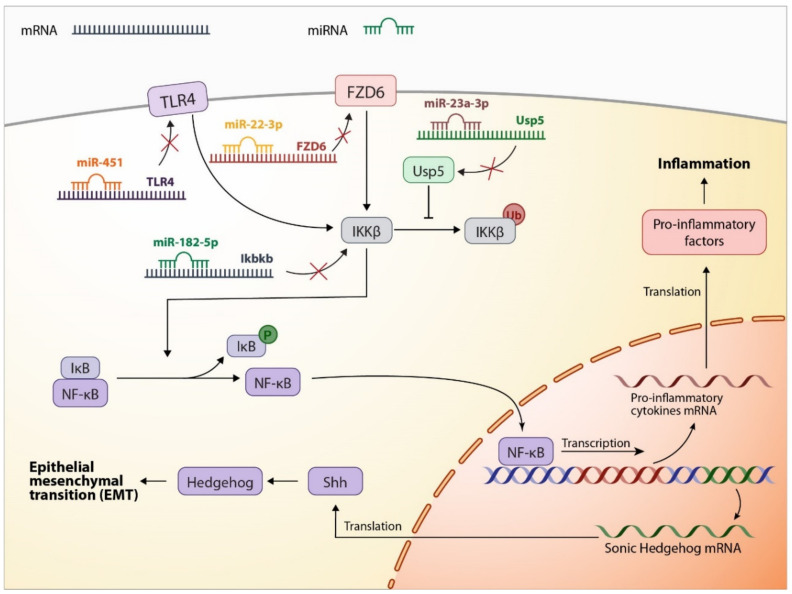
SC-Exo miRNA cargo regulates dysregulated pathways in ARDS. NF-κB is one of the main pathways in which out of balance activity in ARDS leads to lung damage. NF-κB upregulation during ARDS leads to an increased level of proinflammatory factors and sonic hedgehog (Shh) protein, which triggers the hedgehog signaling pathway and begins epithelial–mesenchymal transition in alveolar-epithelial cells. SC-Exos can restore balance to the NF-κB pathway activity and prevent inflammation and EMT, mediated by several miRNAs targeting different proteins upstream of the NF-κB, including TLR4 and IKKβ. FZD6 is another target of SC-Exo miRNA, and its suppression can lead to NF-κB downregulation; however, it is mainly associated with the Wnt/β-Catenin pathway, and the underlying relation between FZD6 and NF-κB is not fully determined.

**Figure 3 ijms-23-06273-f003:**
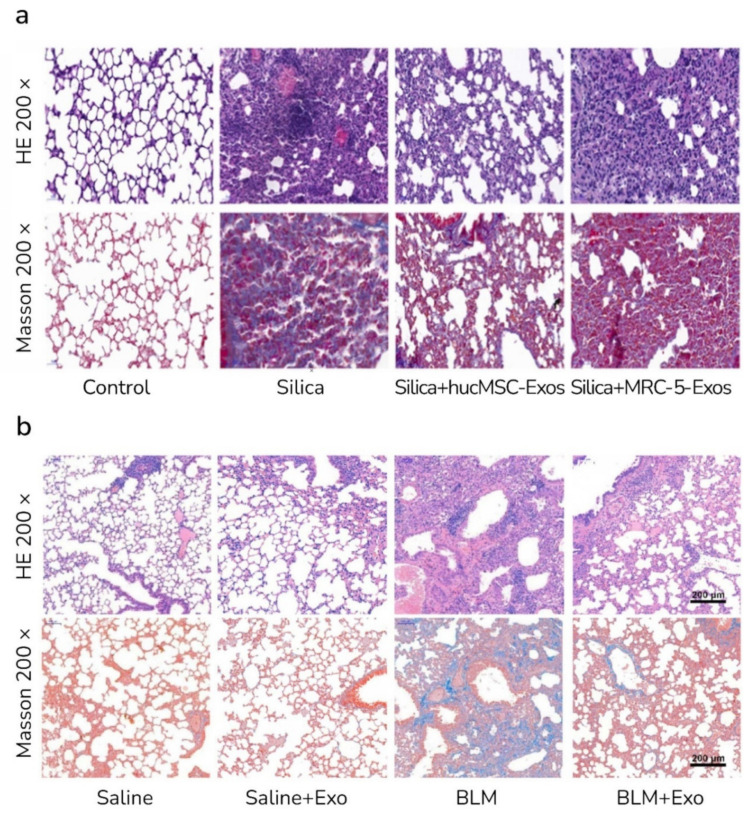
Effects of SC-Exos on PF in mice. (**a**) Human UCMSC-derived exosomes (hucMSC-Exos) but not MRC-5-Exos alleviated silica-induced mice model of pulmonary fibrosis (PF) (H&E and Masson staining; magnifications of 200× via light micrograph [75]. (**b**) Menstrual blood-derived SC-derived exos showed protective effects in bleomycin (BLM)-induced PF in mice (H&E and Masson staining; the scale bar represents 200 μm) [76].

**Figure 4 ijms-23-06273-f004:**
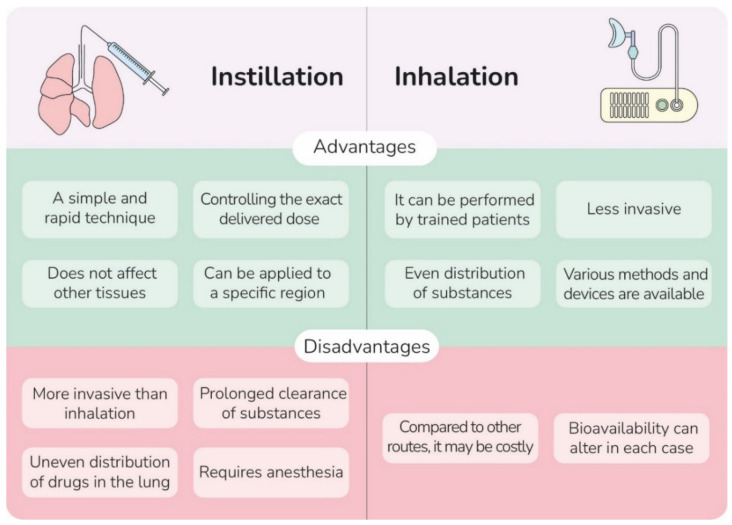
A comparison of advantages and disadvantages of intratracheal instillation and inhalation related to human use.

**Figure 5 ijms-23-06273-f005:**
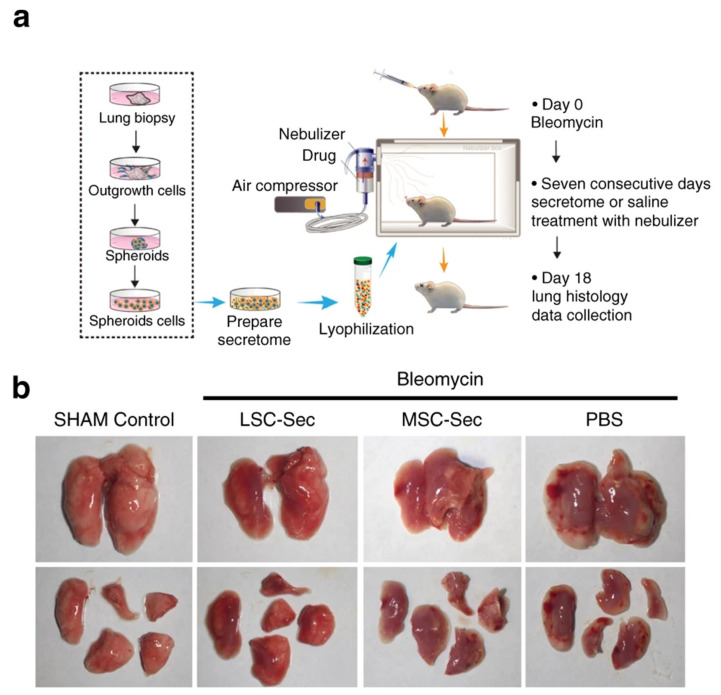
Administration of SC-Exos via the inhalation route. (**a**) The schematic representation of lung spheroid cell (LSC)-secretome (LSC-sec) preparation and animal model treatment with aerosolized secretome. The mice model of PF was obtained by intratracheal injection of bleomycin, and after ten days, 10 mg of secretome protein per kg of body weight was introduced to mice by a nebulizer for seven consecutive days. On day 18, animals were euthanized, and lungs were harvested for histological analysis. (**b**) Macroscopic view of harvested lungs from bleomycin-induced mice. LSC-sec and MSC-sec improved the lung condition in PF mice [77].

**Table 1 ijms-23-06273-t001:** Effects of stem cell-derived exosomes (SC-Exos) on acute respiratory distress syndrome (ARDS) models.

Factor	Study Model	Cell Source	Target	Effect	Reference
miR-182-5p, miR-23a-3p	Lipopolysaccharide (LPS)-induced ARDS	Mesenchymal stem cells (MSCs)	Ikbkb, USP5	Immunomodulation and tissue recovery via IKKβ/nuclear factor kappa-light-chain-enhancer of activated B cells (NF-κB)/hedgehog suppression	[35]
miR-377-3p	LPS-induced ARDS	Umbilical cord-derived MSCs (UCMSCs)	Regulatory associated protein of mammalian target of mTOR (RPTOR)	Injury recovery by RPTOR suppression and inducing autophagy	[36]
miR-22-3p	LPS-induced ARDS	Umbilical cord blood-derived MSCs (UCBMSCs)	Frizzled class receptor 6 (FZD6)/Wnt	Immunomodulation via decreasing FZD6 expression, leading to NF-κB suppression	[37]
miR-150	LPS-induced ARDS	Bone marrow-derived MSCs (BMSCs)	MAPK	Immunomodulation via suppression of MAPK-associated proteins and cleaved caspase-3 and B-cell lymphoma 2 (Bcl-2) promotion	[38]
miR-30b-3p	LPS-induced ARDS	MSCs	SAA3	Immunomodulation via suppression of serum amyloid A-3 (SAA3)	[39]
miR-425	HYRX-induced ARDS	BMSCs	PTEN	PI3K/AKT upregulation via suppressing PTEN	[40]
miR-126	Histone-induced ARDS	Adipose-derived MSCs (AMSCs)	PI3K/AKT	PI3K/AKT upregulation	[41]
Unknown	IIR-induced ARDS	BMSCs	TLR4/NF-κB	Immunomodulation via downregulation of toll-like receptor 4 (TLR4)/NF-κB pathway	[42]
miR-384-5p	LPS-induced ARDS	BMSCs	Beclin-1	Immunomodulation via inducing autophagy in disordered macrophages	[43]
miR-451	Burn-induced ARDS	UCMSCs	TLR4/NF-κB	Immunomodulation via TLR4 and NF-κB downregulation	[31]

**Table 2 ijms-23-06273-t002:** Preclinical studies that used intratracheal instillation for delivering SC-Exos.

Disease	Source of Exosomes	Instilled Dose	Animal Model	Refs
Severe steroid-resistant asthma (SSRA)	Human UCMSCs	100 μg Exos in 50 μL Phosphate-buffered saline (PBS)	Ovalbumin (OVA)/: (complete Freud’s adjuvant) CFA-induced SSRA mice	[89]
Ischemia/reperfusion (I/R) injury	Murine BMSC	Exos derived from 2 × 10^6^ in 30 μL PBS	I/R model mice	[97]
Cigarette smoke (CS)-induced lung injury	Human ASC	30 μL of purified Exos solution (as explained in the original paper)	CS-exposed C57BL/6 mice	[95]
ARDS	BMSC	50 mg Exos in 10 mL PBS and 100 mg Exos in 10 mL PBS	LPS-induced ARDS C57BL/6 mice	[46]
Phosgene-induced acute lung injury (ALI)	Rat BMSC	Exos isolated from 3 × 10^6^ MSC (50 mL)	Phosgene-induced ALI Sprague–Dawley rat	[108]

**Table 3 ijms-23-06273-t003:** Clinical trials conducted with SC-Exos on patients with respiratory diseases (www.clinicaltrials.gov; accessed on 20 April 2022.).

Trail Identification	Official Title	Conditions	Source of Exosomes	Administration Route	Status
NCT04602104	A Clinical Study of Mesenchymal Stem Cell Exosomes Nebulizer for the Treatment of ARDS	ARDS	Human MSCs	Inhalation	Recruiting
NCT04544215	A Clinical Study of Mesenchymal Progenitor Cell Exosomes Nebulizer for the Treatment of Pulmonary Infection	Drug-resistant	Human ASCs	Inhalation	Recruiting
NCT04313647	A Tolerance Clinical Study on Aerosol Inhalation of Mesenchymal Stem Cells Exosomes In Healthy Volunteers	Healthy	MSCs	Inhalation	Completed
NCT04602442	Safety and Efficiency of Method of Exosome Inhalation in COVID-19 Associated Pneumonia	COVID-19 Pneumonia	MSCs	Inhalation	Enrolling by invitation
NCT04491240	Evaluation of Safety and Efficiency of Method of Exosome Inhalation in SARS-CoV-2 Associated Pneumonia.	COVID-19 Pneumonia	MSCs	Inhalation	Completed
NCT04276987	A Pilot Clinical Study on Inhalation of Mesenchymal Stem Cells Exosomes Treating Severe Novel Coronavirus Pneumonia	COVID-19 Pneumonia	Allogenic ASCs	Inhalation	Completed
NCT05216562	Efficacy and Safety of EXOSOME-MSC Therapy to Reduce Hyper-inflammation In Moderate COVID-19 Patients	SARS-CoV2 Infection	MSCs	Intravenous	Recruiting
NCT04798716	The Use of Exosomes for the Treatment of Acute Respiratory Distress Syndrome or Novel Coronavirus Pneumonia Caused by COVID-19	COVID-19 ARDS	MSCs	Intravenous	Not yet recruiting

## Data Availability

Not applicable.

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
