# Peer review of "Molecular Insight into the Therapeutic Effects of Stem Cell-Derived Exosomes in Respiratory Diseases and the Potential for Pulmonary Delivery"

_ijms, 2022, doi:10.3390/ijms23116273_

Round 1
Reviewer 1 Report
This is a really nice review. It is of direct interest to this journal and is well written. There are some trivial things that need correcting before it is published however these do not detract from he manuscripts value once corrected. They are: Line 80. Is the word administration missing after parental ? Line 88-94. This needs clarifying. MVB are synonymous with late endoscopes (LAMP positive, M6PR positive structures). Line 105-106 the sentence seems to wander off and suddenly stop. Perhaps making reference to https://doi.org/10.1083/jcb.200312072 might be useful ? Line 184, is the capital B on bone intentional ? Line 209, Is some explanation as to the nature of Worton's Jelly ? There are a couple of in vivo and in vitro that could probably be with italicising. Nice job !
Author Response
Manuscript ID: ijms-1728437
Response to Reviewer
Dear editors,
Thank you very much for having considered our manuscript " Molecular insight into the therapeutic effects of stem cell-derived exosomes in respiratory diseases, and the potential for pulmonary delivery" for publication in the Journal of International Journal of Molecular Sciences. We appreciate to have received a positive evaluation, and we would like to express our appreciation to the Reviewers for the thoughtful comments and helpful suggestions. We have incorporated all of the suggestions made by the reviewers. Our detailed, point-by-point responses in blue to reviewer comments are given below, whereas the corresponding revisions are marked by using Track Changes in the manuscript file.
Response to Reviewer #1
This is a really nice review. It is of direct interest to this journal and is well written. There are some trivial things that need correcting before it is published however these do not detract from he manuscripts value once corrected. They are:
Author response: Thank you for your positive feedback. We really appreciate you taking the time to review this paper.
Point 1: Line 80. Is the word administration missing after parental?
Author response: We thank the reviewer for pointing out this issue which was revised in the manuscript.
Point 2: Line 88-94. This needs clarifying. MVB are synonymous with late endoscopes (LAMP positive, M6PR positive structures).
Author response: We thank the reviewer for pointing out this issue which was revised in the manuscript as below:
"Exosome biogenesis starts with the formation of an early endosome that forms by several endocytic vesicles fusion. Exosomal cargos are then segregated into microdomains on the endosome membrane. Further, these microdomains bud into the lumen of the endosome and form a multivesicular body (MVB), also referred as late endosome, containing numerous intraluminal vesicles (ILVs) that can be secreted upon fusion of this MVB with the plasma membrane or become degraded by MVB fusion with the lysosome [18]."
Point 3: Line 105-106 the sentence seems to wander off and suddenly stop. Perhaps making reference to https://doi.org/10.1083/jcb.200312072 might be useful?
Author response: To address this issue, we revised the statement which is shown below.
“In addition, non -ubiquitinated proteins can be sorted into ILVs via the ESCRT pathway intersection by syntenin and ALIX (ALG-2-interacting protein X; an ESCRT and bridge them and the Vps32 (Vacuolar sorting-associated protein 32) ESCRT-III subunit, respectively.”
Point 4: Line 184, is the capital B on bone intentional?
Author response: The typo was revised in the manuscript
Point 5: Line 209, Is some explanation as to the nature of Worton's Jelly?
Author response: Very short explanation was added to the manuscript
Point 6: There are a couple of in vivo and in vitro that could probably be with italicising.
Author response: We thank the reviewer for pointing this out. We have incorporated your suggestion throughout the manuscript.

Reviewer 2 Report
Mohammad H. Azhdari and colleagues summarized the molecular mechanism of using stem cell (SC) derived exosomes in respiratory diseases. They highlighted the biogenesis of exosomes, and their applications in several lung diseases, including acute respiratory distress syndrome, bronchopulmonary dysplasia, pulmonary hypertension, pulmonary fibrosis, asthma and other pulmonary disorders. They also discussed the delivery routes and the clinical trials of SC-derived exosomes against lung diseases. The topic is important and the paper is well written. Here are some suggestions and concerns:
1. There are a few typos, for example in Table 1, 2nd row, “vsuppression” should be corrected. Please double check the spell of “ubiquitination” in line 143 and “regulates” In line 173.
2. Please add the full name of UMSC in line 144, and full name of MRC in line 324.
3. Why the “Neutralizing” is capitalized in line 249?
4. Do you mean five subtypes, instead of four, in line 257?
5. In line 355, please confirm the scale bar size, which was labels as 200 um in figures and 5 mm in the figure legend.
6. In Table 2, 3rd row, please add the absolute amount (ug or mg) of exosomes used in reference 95.
Author Response
Manuscript ID: ijms-1728437
Response to Reviewer
Dear editors,
Thank you very much for having considered our manuscript " Molecular insight into the therapeutic effects of stem cell-derived exosomes in respiratory diseases, and the potential for pulmonary delivery" for publication in the Journal of International Journal of Molecular Sciences. We appreciate to have received a positive evaluation, and we would like to express our appreciation to the Reviewers for the thoughtful comments and helpful suggestions. We have incorporated all of the suggestions made by the reviewers. Our detailed, point-by-point responses in blue to reviewer comments are given below, whereas the corresponding revisions are marked by using Track Changes in the manuscript file.
Response to Reviewer #2
Mohammad H. Azhdari and colleagues summarized the molecular mechanism of using stem cell (SC) derived exosomes in respiratory diseases. They highlighted the biogenesis of exosomes, and their applications in several lung diseases, including acute respiratory distress syndrome, bronchopulmonary dysplasia, pulmonary hypertension, pulmonary fibrosis, asthma and other pulmonary disorders. They also discussed the delivery routes and the clinical trials of SC-derived exosomes against lung diseases. The topic is important and the paper is well written. Here are some suggestions and concerns:
Author response: Thank you for your positive feedback. We really appreciate you taking the time to review this paper.
Point 1: There are a few typos, for example in Table 1, 2nd row, “vsuppression” should be corrected. Please double check the spell of “ubiquitination” in line 143 and “regulates” In line 173.
Author response: Thank you for pointing these out. We have incorporated your suggestion throughout the manuscript.
Point 2: Please add the full name of UMSC in line 144, and full name of MRC in line 324.
Author response: The full name of MRC was added, and also changed UMSC into UCMSC which its full name was mentioned in the manuscript in table 1 row 2.
Point 3: Why the “Neutralizing” is capitalized in line 249?
Author response: Thank you for pointing out this typo. It was revised in the manuscript.
Point 4: Do you mean five subtypes, instead of four, in line 257?
Author response: This issue was revised in the manuscript. five subtypes was replaced for four subtypes.
Point 5: In line 355, please confirm the scale bar size, which was labels as 200 um in figures and 5 mm in the figure legend.
Author response: Thank you for pointing it out. 200 µm was substituted for 5 mm.
Point 6: In Table 2, 3rd row, please add the absolute amount (ug or mg) of exosomes used in reference 95.
Author response: We thank the reviewer for pointing this out, but we are afraid the reference paper authors did not mention the absolute number, and only explained the procedure of making the solution containing exosomes.
